# A Review and Perspective for the Development of Triboelectric Nanogenerator (TENG)-Based Self-Powered Neuroprosthetics

**DOI:** 10.3390/mi11090865

**Published:** 2020-09-18

**Authors:** Hao Wang, Tianzhun Wu, Qi Zeng, Chengkuo Lee

**Affiliations:** 1Institute of Biomedical & Health Engineering, Shenzhen Institutes of Advanced Technology (SIAT), Chinese Academy of Sciences (CAS), Shenzhen 518035, China; tz.wu@siat.ac.cn (T.W.); qi.zeng@siat.ac.cn (Q.Z.); 2Key Laboratory of Health Bioinformatics, Chinese Academy of Sciences, Shenzhen 518035, China; 3Department of Electrical and Computer Engineering, National University of Singapore, Singapore 117576, Singapore; elelc@nus.edu.sg

**Keywords:** triboelectric nanogenerator (TENG), self-powered neuroprosthetics, thin-film, Bennet’s doubler, high-frequency switch, implantable device

## Abstract

Neuroprosthetics have become a powerful toolkit for clinical interventions of various diseases that affect the central nervous or peripheral nervous systems, such as deep brain stimulation (DBS), functional electrical stimulation (FES), and vagus nerve stimulation (VNS), by electrically stimulating different neuronal structures. To prolong the lifetime of implanted devices, researchers have developed power sources with different approaches. Among them, the triboelectric nanogenerator (TENG) is the only one to achieve direct nerve stimulations, showing great potential in the realization of a self-powered neuroprosthetic system in the future. In this review, the current development and progress of the TENG-based stimulation of various kinds of nervous systems are systematically summarized. Then, based on the requirements of the neuroprosthetic system in a real application and the development of current techniques, a perspective of a more sophisticated neuroprosthetic system is proposed, which includes components of a thin-film TENG device with a biocompatible package, an amplification circuit to enhance the output, and a self-powered high-frequency switch to generate high-frequency current pulses for nerve stimulations. Then, we review and evaluate the recent development and progress of each part.

## 1. Introduction

Neuroprosthetics is a powerful toolkit, for clinical interventions of various diseases, affecting the central nervous or peripheral nervous systems by electrically stimulating different neuronal structures [1]. Deep brain stimulation (DBS), based on the electrical stimulation of deep structures within the brain, is clinically used for symptomatic treatment of motor-related disorders, such as Parkinson’s disease, dystonia, and tremor, and it is also under clinical development for other drug-resistant neurological disorders, such as depression, obsessive-compulsive disorder, and others [2]. Electrical stimulation of the central nervous system (CNS) and peripheral nervous system (PNS) can also be achieved by implanted neuroprosthetic devices in the spinal cord or peripheral nerves and muscles to restore sensory and motor function in a novel and promising field of therapeutic interventions termed “bioelectronics” [3,4,5,6,7]. To prolong the lifetime of the implanted devices, researchers have invested enormous enthusiasm and energy to develop the power sources for them [8,9,10,11]. One of the primary strategies is harvesting power from organs or the surrounding environment, which is also called a self-powered method. Currently, there are six major research directions for achieving this self-powered method by harvesting energy from our own body. Triboelectric nanogenerators (TENGs) and piezoelectric nanogenerators harvest energy from the motion of organs; optical devices, ultrasound devices, and electromagnetic coils receive energy wirelessly by a transcutaneous approach; and biofuel cells extract energy from the redox reaction of glucose and electrolytes in the gastrointestinal tract [12]. Among all these methods, a TENG is the onlymethod to achieve direct nerve stimulations [13,14,15,16,17,18,19,20,21], showing its capability in the realization of a self-powered neuroprosthetic system in the future, which is far beyond the application of energy harvesting and self-powered physical and chemical sensing [22,23,24,25,26,27,28,29,30,31,32,33,34,35,36,37,38,39,40,41,42,43,44,45,46,47,48,49,50,51,52,53,54,55].

In this review, the current development and progress of TENG-based stimulations of various nervous systems are systematically summarized. Then, based on the requirements of the neuroprosthetic system in a real application and the development of current techniques, a perspective of a more sophisticated neuroprosthetic system is proposed, including components of the thin-film TENG device with a biocompatible package, an amplification circuit to enhance the output, and a self-powered high-frequency switch to generate high-frequency current pulses for nerve stimulations. Then, the recent development and progress of each part are reviewed and evaluated.

## 2. The Development of Triboelectric Nanogenerator (TENG)-Based Self-Powered Nerve Stimulation

The feasibility of nerve stimulations by a TENG was firstly validated by Zhang et al., in 2014 [13]. The TENG device of a folding structure (Figure 1(A1)) was connected with a three-dimensional (3D) microneedle electrode array (MEA) (Figure 1(A2)), which was implanted into real frog tissue to stimulate the sciatic nerve. When the external force was applied to the TENG device, the instantaneous output voltage induced the loop current among the microneedle tips via the sciatic nerve. Therefore, the sciatic nerve was stimulated by the loop current and actuated the leg muscle of the frog, as shown in Figure 1(A3).

In 2018, Yao et al. presented a TENG-based self-powered vagus nerve stimulation (VNS) system on a rat (Figure 1B) [14] for the application of weight control. A flexible TENG device was attached to the stomach surface to generate alternating pulses from the stomach peristalsis (Figure 1(B1-i)). As the expansion and contraction of the stomach, two triboelectric layers were driven to the contact-separation cycle, generating the current pulse required for nerve stimulations (Figure 1(B2)). The electrical output was conducted through two gold leads to the proximity of the gastroesophageal junction around the anterior vagus nerve (AVN) and posterior vagus nerve (PVN). With the TENG-based VNS, the rats always had less food intake, and thus it was easier to lose weight or gain less weight.

Lee et al. conducted a series of studies on TENG-based stimulations of various kinds of nervous systems, including the sciatic nerve and branches [15,16], pelvic nerve [17], and muscle nerves [18,19,20,21].

For the first validation of the sciatic nerve stimulation on a rat, a stack-layer TENG device is prepared (Figure 1(C1)) and connected with a neural electrode, called a sling electrode [16]. This sling electrode wraps around the sciatic nerve with a tight contact with the nerve bundle (Figure 1(C2)). With a 2 Hz tapping frequency, the TENG device generates current pulses of around 1 µA to stimulate the sciatic nerve. This current amplitude is insufficient to directly activate a visible leg movement, therefore, the electromyography (EMG) signal of the gastrocnemius medialis and tibialis anterior muscles was recorded to confirm the effectiveness of the sciatic nerve stimulation (Figure 1(C3)). Although 1 µA current was not enough to induce a leg movement by directly stimulating the sciatic nerve, it was sufficient to achieve leg kicking by stimulating the nerve branches in the sciatic nerve. There are three nerve branches in the sciatic nerve; among them, the tibial nerve and common peroneal nerve control the tibialis anterior muscle and gastrocnemius medialis muscle. Thus, it is possible to separately stimulate the tibial nerve and peroneal nerve to achieve ankle dorsiflexion (forward kicking) and plantarflexion (backward kicking). Lee et al. developed a multi-pixel TENG device to achieve this selective stimulation (Figure 1D) [15]. Two different pixels were connected with the Pt/Ir wires wrapping around the tibial nerve and peroneal nerve, respectively. Each pixel was 2 × 2 cm in dimension, and the output current was about 0.7 µA. By tapping one of the pixels, which was connected to either the tibial nerve or peroneal nerve, a visible plantarflexion or ankle dorsiflexion was observed (Figure 1(D2,D3)). Lee et al. also achieved the stimulation of a rat’s autonomic pelvic nerve by a similar TENG device, shown in Figure 1(C1) (4 × 4 cm with four layers) to control the bladder pressure and micturition [17]. The required current was about 6 µA to activate the autonomic pelvic nerve and induce bladder contraction. By controlling the times and frequency of the tapping of the device, the bladder contraction could be effectively controlled. When the tapping frequency was more than 50 Hz and the tapping times were more than once, a visible bladder contraction with micturition was observed. These results show that a small dimension TENG device (not larger than 4 × 4 cm) can effectively provide the current pulse for nerve stimulations.

Electrical muscle stimulation is a critical clinical therapy for the treatment of muscle function loss induced by spinal cord injury (SCI), stroke, chronic venous insufficiency (CVI), and multiple sclerosis. Thus, TENG-based muscle stimulation is a potential solution. However, the challenge of directly stimulating a muscle using a TENG device is obvious. Different from peripheral nerves such as sciatic nerves and autonomic pelvic nerve, which are more spatially concentrated in a nerve bundle, excitable motoneurons are sparsely distributed in the muscle tissue, as shown in Figure 2(A1). A neural probe inserted into a muscle cannot have close contact with these motoneurons. Therefore, the threshold current required for muscle stimulation is normally 10 times higher than that of the sciatic nerve stimulations. Wang et al. used a stack-layer TENG with a larger size (Figure 2(A1)) to enhance a short circuit current to about 35 µA for muscle stimulation (Figure 2(A2i)) [18,19]. The neural probe used for muscle stimulation was an intramuscular neural probe with six pads (Figure 2(A3i)). Since there were six pads and only two pads were selected and connected to the TENG device for the muscle stimulation, there were 30 possible combinations. Different electrode pad combinations had different stimulation efficiencies due to their various distances to the target motoneurons. Even with an enhanced output of 35 µA, only several specific combinations could achieve muscle stimulation (Figure 2(A4)). The measured force was about 120 mN. Thus, this result indicated that the current generated by the TENG device was not high enough for a stable and high-efficiency muscle stimulation. Therefore, Wang et al. further developed a switch which involved an output method to further enhance the output current (Figure 2B) [20]. The nature of the TENG device was a self-charged capacitor. Therefore, the generated energy by the TENG actually could be accumulated if the external circuit was opened by a switch. Then, when the switch was closed, all the accumulated energy was discharged immediately, which was an RC discharge, generating a current pulse with a very high amplitude (Figure 2(B1ii)). Different from the conventional TENG devices whose current/voltage amplitude is determined by the tapping speed, the current amplitude of this switchable TENG device was determined by the equation *I* = *V*/*R*. Here, *V* was the open-circuit voltage of the TENG device, and *R* was the external load resistance. In this study, the device had the same structure as the one in Figure 2(A1), whose open-circuit voltage was about 50 V and impedance of the neural tissue was about 10 KΩ, the current amplitude of the RC discharge could be 5 mA, which was much larger than the previous 35 µA. Therefore, with the switching method, the force generated by the muscle stimulation was larger (Figure 2(B2)). Meanwhile, all electrode pad combinations could achieve stable muscle stimulations (Figure 2(B3)). Since this switching method significantly enhanced the current amplitude of the TENG device, it was possible to further reduce the number of layers and size of TENG device. Therefore, He et al. developed a fabric-based thin-film TENG device that could also achieve stable and effective muscle stimulation (Figure 2C) [21]. Conventionally, thin and soft TENGs cannot generate high output. However, with the switching method, even the thin film fabric device can generate sufficient current for muscle stimulation. In this study, two intramuscular neural probes were implanted in the anterior tibialis muscle and gastrocnemius muscle, respectively. Each probe was connected to the TENG device through an individual switch. Then, the ankle dorsiflexion and plantarflexion were realized by controlling the switching operation, as shown in Figure 2(C2). The control mechanism is a unique feature of this fabric-based thin-film TENG device, which is also desired for functional electrical stimulation. The actual output current was proportional to the pressing area (Figure 2(C3)). Meanwhile, a strength of this device was that the muscle stimulation was also proportional to the amplitude of the generated current. Therefore, the force generated could be effectively controlled by the pressing area on the fabric-based thin-film TENG device.

In summary, the feasibility of nerve and muscle stimulation by a TENG device of a limited dimension has been validated. Now, even a thin film device can generate a current pulse higher than 100 µA, which is sufficient for the electrical stimulation of almost all kinds of nervous systems. For the next step, a more sophisticated implantable neuroprosthetic system should be developed.

## 3. The Perspective for a TENG-Based Self-Powered Neuroprosthetic System

Although Yao et al. demonstrated a self-powered VNS system on a rat for a weight control application [14], this system could hardly meet the requirement for most neural modulation applications such as deep brain stimulation (DBS), vagus nerve stimulation (VNS), and functional electrical stimulation (FES) in humans. This is due to the fact that the current pulse frequency of TENG devices attached to the stomach and heart cannot be higher than 2 Hz, but the frequency of the current pulses required for a neuroprosthetic system is typically higher than 10 Hz. Therefore, the current demonstrated TENG-based neuroprosthetic system should be improved. Here, based on the state of the ongoing development of TENG-based nerve stimulation and relevant techniques, a perspective of the TENG-based self-powered neuroprosthetic system is proposed.

This TENG-based self-powered neuroprosthetic system consists of four major components, as shown in Figure 3. The first component is a thin film implantable TENG device, which requires a biocompatible package and can be driven by the heart, stomach, or hand tapping, if implanted beneath the skin. Since the frequency of the mechanical operation cannot fulfil the high-frequency current pulses required for neural modulation, a high-frequency switch is necessary to generate high-frequency current pulses (component 3 in Figure 3). The operation of this high-frequency switch can easily be achieved by a field-effect transistor (FET) or a relay, which is explained in detail in a subsequent section. However, when a single current pulse of low frequency is divided into a series of current pulses of high frequency, the amplitude of each current pulse decreases, and thus can be insufficient for nerve stimulations. Therefore, between components 1 and 3, an amplification circuit is required to amplify the output of the TENG device and ensure that each current pulse is high enough for nerve stimulations (component 2 in Figure 3). Then, the last component is the neural interface to enable the electrical nerve stimulation.

## 4. The Development of TENG-Based Self-Powered Nerve Stimulation

### 4.1. The Development of Thin-Film TENG Devices for Implantable Applications

Figure 4 shows the general working mechanism of a TENG device with the contact-separation operation. The triboelectrification effect appears at the interface between two different contacted dielectrics. When the two friction layers are brought into contact, triboelectric charges are transferred to the surfaces of the two, due to the triboelectrification effect. Once the contact of the friction layers is finished, they can carry the same number of opposite signs of triboelectric charges. When the two friction layers are separated, there is an electrostatic field build by the triboelectric charges, which can drive the electrons to flow through the external load. Apparently, the output power is determined by the number of generated triboelectric charges, which is proportional to the dimension of the device. This is the reason that TENG devices for harvesting energy from wind or ocean waves are normally designed with a large size [56,57,58,59,60,61]. However, the thin-film TENG devices used for implantable applications meet more constraints.

The first constraint is the mechanical design. It has been shown that a successful working cycle of a TENG device relies on a stable contact-separation operation, which requires an operation spacing and also a mechanism to separate the two layers after the pressing. How to achieve this operation spacing and the automatic releasing mechanism is the key to designing a thin film TENG device. Currently, there are two possible designs to achieve this automatic releasing mechanism. The first design is the spacer structure, as shown in Figure 4(B1). Between the two friction layers, there is a spacer to introduce the gap. To ensure the two layers can separate after the pressing, i.e., the two friction layers, there should be a backbone made with highly resilient materials, such as titanium. Then, considering that there are also other layers used for friction, electrode, and encapsulation, the actual thin-film device will have a multi-layer structure, as shown in Figure 4(B1) [62]. Another design is the “keel structure”, as shown in Figure 4(B2). The curved keel structure made of titanium can guarantee the contact-separation process [63]. Apparently, such a spacer or keel structure can only work when the gap is very small. However, a larger gap is always desired for enhancing the output power. Therefore, a further optimized design can be a combination of a spacer and a keel structure, as shown in Figure 4(B3)) [64], which achieves a larger gap and generates higher output. Considering that the above-mentioned three designs share similar materials and encapsulation process, their open-circuit voltage, *V*_oc_, can be a fair comparison. The design combining spacer and keel structure has a *V*_oc_ of 90 V [64], which is higher than that of spacer design, i.e., 20 V [62], and keel structure design, i.e., 60 V [63].

Another constraint for thin-film TENG devices is the power source. The speed of the mechanical operation also determines the amplitude of the output voltage/current. Since the total quantity of the transferred charge is constant, a faster contact-separation process can squeeze the charge transfer process within a shorter duration, and thus generate a pulse with higher amplitude. Therefore, the power source to press/squeeze thin-film TENG devices should have a quick and rhythmic mechanical movement. Currently, it seems that only the heart beating [63] and external manual pressing can fulfill this requirement. The mechanical movement of other organs cannot effectively drive the thin-film TENG device. For example, stomach peristalsis typically has a very low frequency of 0.05 Hz. Therefore, in the demonstration of VNS with the TENG device attached to the stomach surface, the output voltage is around 0.1 V [14], which is much lower than the one attached to the heart [63].

Recently a switchable TENG device has been proposed, whose characteristic is fundamentally different from a conventional TENG device. Its application in a thin-film TENG device can help overcome the two constraints mentioned above. The primary operating mechanism of the switchable TENG device is shown in Figure 5A. An extra switch is added in the circuit, as shown in Figure 5(A1). Initially, this switch is open, making the whole circuit open. When the two friction layers are separated, the electrostatic charge on the dielectric surface gradually polarizes the electrode (Figure 5(A1ii)). Since the circuit is open, the polarization introduces a charge accumulation around the switch. Closing the switch induces an instantaneous current flow (Figure 5(A1iii)). Then, open the switch and press the device. The charges on the dielectric surfaces neutralize each other and induce a polarization on the electrodes again (Figure 5(A1v)). Closing the switch induces another instantaneous current flow (Figure 5(A1vi)). As seen, the function of this switch is to squeeze the time window of the current flow, which can significantly enhance the amplitude of the measured current. Meanwhile, at the instant of closing the switch, the TENG device can be modeled as a charged capacitor whose capacitance is a constant. Closing the switch induces an RC discharge, whose waveform follows an exponential curve, which is different from the voltage waveform of a conventional TENG device (Figure 5(A2)). This switching operation can be realized by physical contact of two pieces of metal, which is easily integrated within any mechanical design. For example, in a sliding mode TENG device, as shown in Figure 5B [65], some electrode pads can be arranged at certain positions to achieve this switching operation automatically in the function of the device (Figure 5(B-ii)). One of the key advantages of involving this switching operation in the TENG device is the upboosting of the output power and modification of the inner impedance. Conventionally, the capacitive nature of the TENG device introduces a huge inner impedance, which is normally at the MΩ level. This huge inner impedance cannot directly match with the impedance of normal electronics. This is the reason that a TENG device cannot directly power an electric device but has to store the generated energy in a power management unit (PMU) first [66,67,68]. However, the switching operation changes the output mode of the TENG device to an RC discharging pattern, which has no inner impedance. The output power diverges when the load resistance approaches zero (Figure 5(B-iii)) (TENG-UDS refers to TENG with an unidirectional switch, whereas TENG-WOS refers to TENG without switch). This switching operation is also adopted in the power management method, as shown in Figure 5C [69]. The charge/energy accumulation/releasing by the switching operation is a universal method that helps to maximize the energy harvesting efficiency and enables a more sophisticated power management strategy. For example, based on the switching operation, a controlled LC oscillating circuit composed of a diode and inductor can help to further enhance the output energy and enable the impedance matching with low-impedance common electronics (Figure 5D) [70].

In conventional TENG devices, a faster mechanical operation can achieve a higher output. However, by introducing the switching operation, the operation speed does not affect the amplitude of the output anymore. This is because the switching operation makes the charge/energy generation and release to two independent processes. This unique feature enables the feasibility of soft TENG devices, as shown in Figure 5E [71]. This soft thin-film TENG device consists of Nitrile rubber and EcoFlex^TM^ as two friction layers, and two pieces of conductive tapes or textiles as electrodes and supporting layers. The curved Nitrile rubber introduces a small spacer, which is equivalent to the small gap in normal TENG devices. On the one hand, due to its soft nature, it cannot achieve an instantaneous contact-separation process, and thus cannot generate a very high output at a non-open-circuit condition, which is its key disadvantage. On the other hand, its soft nature enables the sensing of the pressing area and force. Therefore, similar soft devices are widely used for all kinds of self-powered physical sensing, such as foot stepping, hand gesture and body motion, in wearable electronics [72,73,74,75]. However, integration of a soft thin-film TENG device and a switch changes the whole story. The switching operation can release the charge/energy generated by the slow and gradual contact-separation process immediately and forms an RC discharging voltage of an exponential waveform, whose time constant is determined by the instantaneous capacitance of the TENG device. According to this time constant, the force or the bending applied upon the TENG device can be further decoded, which is a kind of force-sensing mechanism (Figure 5(E2i–iii)). This device can be attached to the elbow or beneath the foot for sensing the bending angle or the stepping force (Figure 5(E2vi)) [75]. The RC discharge induced by the switching operation surely can be extended to the application of wireless transmission [76]. Just with an inductor connected in series to form an RLC circuit, the switching operation can generate an oscillation and emit an electromagnetic wave, whose frequency is determined by the equation $1/2\pi \sqrt{LC}$. Since the inductance is a constant, whereas the capacitance is a controllable variable determined by either the applied force or the design of difference pixels, this device can realize self-powered sensing with the wireless transmission function (Figure 5F). Another unique application for this switchable soft-thin film TENG device is the self-powered nerve/muscle stimulation, as shown in Figure 2C. The soft and thin nature of this TENG device is potentially suitable for implantable applications. Meanwhile, different from the devices with a resilient backbone or keel structure, as shown in Figure 4B, which can only harvest energy from heart, this switchable soft thin-film device can also convert the energy harvested by organs with slow and low frequency movement, such as enterogastric peristalsis, to current pulses of sufficiently high amplitude for nerve/muscle stimulations.

### 4.2. Biocompatible Packages

One of the most critical prerequisites for the implantable device is the biocompatible packages. Currently, two major research groups are exploring the packaging methods of the implantable TENG devices. One is Prof. Zhonglin Wang’s research group. In 2014, they first demonstrated a thin-film TENG device encapsulated with polydimethylsiloxane (PDMS) buried under the thoracic skin for harvesting energy from breathing [77]. In 2016, an improved package strategy was proposed, as shown in Figure 4(B2) [63]. It was a layer-by-layer encapsulation strategy. A core encapsulation layer of polytetrafluoroethylene (PTFE) film (50 µm) was used to ensure the biocompatibility and corrosion resistance. A flexible polydimethylsiloxane (PDMS) layer (200 μm) covered the entire device as the shell package by spin coating to enhance the leak-proof performance and the structural stability of the entire device. To further increase the in vivo reliability of the device and to avoid potential erosion and adhesion in the physiological environment, parylene-C was deposited onto the surface as another shell structure of the device to form a high-density and hole-free coating layer. This package strategy ensured structural stability and its resistance to a complex external environment. This PTFE-PDMS-Parylene-C package strategy was also applied in their later study of implantable TENG devices used for the sensing of the endocardial press [78] and multiple physiological and pathological signs [79].

Additionally, Wang et al.’s research group studied harvesting breathing energy using an implantable TENG device. The device was encapsulated by ecoflex, which was a soft and biocompatible material [80]. In another work of VNS shown in Figure 1B, they also adopted a layer-by-layer strategy [14]. The device was encapsulated by Polyimide (Pi), PDMS, and ecoflex. It seems that this layer-by-layer strategy was more sophisticated for implantable TENG devices. A similar approach using parylene-C/Al_2_O_3_ achieved an excellent hermetical level, up to 10^−10^ Pa·m/s, for a 10-year implant which met the Food and Drug Administration (FDA) medical standard [81].

### 4.3. Circuit to Amplify the Output of TENG Devices

As explained in the perspective of the self-powered neuroprosthetic system, the low-frequency output of the TENG device should be converted to high-frequency current pulses for nerve/muscle stimulations. To ensure that the amplitude of each current pulse is sufficient to activate the nerves, the output of the TENG should be further amplified. Conventionally, the output of the TENG device can be enhanced from two directions. One is surface treatment to increase the surface area. The friction surface can be etched to the rough surface with nanostructures, which significantly increases the surface area, and thus enhances the output [82,83,84]. Another direction is to enhance the surface charge density by corona discharging [85] or introducing electret materials [86,87,88]. All these methods are directly applied to TENG devices. Apart from the TENG device itself, the external circuit can also achieve power amplification. Currently, the most attractive approach is to leverage the Bennet’s doubler conditioning circuit to achieve an exponential enhancement of the output energy [89,90,91,92].

The detailed mechanism of the Bennet’s doubler is shown in Figure 6(A1). The Bennet’s doubler is simple and smart for the continuous doubling of a small initial charge through a sequence of operations with three electrodes. It was invented in 1787 by the Reverend Abraham Bennet, who used it for studies about the electrical state of the air [93]. The original version of the Bennet’s doubler only had three electrodes (A, B, and C in Figure 6(A1)). Wang et al. incorporated an extra electrode (D in Figure 6(A1)) for achieving continuous power output without charge consumption of the whole system [94]. This working mechanism can be physically realized by a mechanical sliding device, shown in Figure 6(A2), to easily achieve very high output voltage (~1.5 kV) with any kind of materials used for friction surfaces (Figure 6(A3)). The charge accumulation by the operation of Bennet’s doubler can also be realized by a Bennet’s doubler conditioning circuit, as shown in Figure 6(B1-i) [90], which only consists of two capacitors and three diodes. During the mechanical operation of the TENG device, the diodes, as automatic switches, can reconfigure the charge storing capacitors between series and parallel modes, as shown in Figure 6(B1iii,iv). In this process, the voltage on the reservoir capacitor ($C_{res}$) keeps increasing (Figure 6(B2)). Figure 6(Bii) shows the QV diagram in the ith cycle of the operation. The advantage of this Bennet’s doubler conditioning circuit on the output power enhancement is obvious. It can be applied to almost any TENG device. Meanwhile, the output voltage keeps increasing during the operation (until the breakdown voltage of the capacitor, diode, and the TENG device). Therefore, the ultimate voltage is only slightly affected by the friction materials of the TENG device. This offers more flexibility of material choices for the implantable TENG device. However, there are two critical requirements for this Bennet’s doubler conditioning circuit to achieve exponential enhancement of the output energy as follows:$$C_{res}\gg C_{store}\gg C_{TENG-Max}$$
$$C_{TENG-Max}>C_{TENG-Min}$$
where $C_{TENG-Max}$ refers to the maximum capacitance of the TENG device when it is fully pressed, and $C_{TENG-Min}$ refers to the minimum capacitance of the TENG device when it is fully separated.

A more detailed circuit analysis for this Bennet’s doubler conditioning circuit can be found in the study by Ghaffarinejad et al. [90]. The first requirement can be easily fulfilled since we can select the capacitors with proper parameters. However, the second requirement is a challenge to the thin-film TENG devices. The spacing of the gap is the main factor in determining the changing ratio of the capacitance of the TENG device. The Bennet’s doubler conditioning circuit prefers a larger gap, while the implantable application restricts the thickness. Therefore, for the next step, developing a thin-film TENG device with higher capacitance changing ratio is the key milestone.

### 4.4. High-Frequency Switch for Current Pulse Generation

A high-frequency switch is required to generate high-frequency current pulses for continuous nerve/muscle stimulations. Considering the whole system should be self-powered, the switching mechanism should also be operated without an external power supply. Currently, there are several automatic switching methods that can be applied to TENG devices, as shown in Figure 7. The first one is the plasma-based switch [95,96]. A tip-electrode structure can be used as a switch, which generates current pulses when the voltage from the TENG device exceeds the threshold of air breakdown (Figure 7(A1)). This air breakdown is measured as a series of current pulses, as shown in Figure 7(A2). This plasma initiated from the electrode tip is less affected by the distance between the tip and electrode. However, the plasma initiated from the electrode is affected by this tip-electrode distance, as shown in Figure 7(A3). The disadvantage of this method is that the threshold voltage required for the switching process is too high (normally at hundreds of volts). The energy of each current pulse is too high for safe nerve stimulations. Meanwhile, an implantable device can only generate limited power, which cannot induce multiple current pulses in one operation cycle. Another method is an electrostatic automatic switch, as shown in Figure 7(B1) [97,98]. This is a follow-up research by Ghaffarinejad et al. for further developing the energy harvesting system based on the Bennet’s doubler condition circuit [90]. This electrostatic automatic switch is constructed using a copper wire as the mobile electrode, a fixed aluminum plate, and a micro-positioning platform to control the gap between the two electrodes. The voltage generated by the TENG device induces electrostatic force between the copper wire and the aluminum plate, and thus pulls in the copper wire and releases the accumulated energy. The advantage of this switching method is that the threshold voltage can be controlled by the gap distance. However, unless a microelectromechanical system (MEMS)-based micro-switch is developed, this macro-scale switch cannot reduce the threshold voltage to the range of several volts (Figure 7(B2)). This macro-scale switch is an exquisite setup that is vulnerable to disturbance and not suitable for integration with implantable devices. A more widely applied switching method is MOSFET. However, a metal-oxide-semiconductor field-effect transistor (MOSFET) is an active component that requires a gating voltage. Therefore, a circuit, as shown in Figure 7(Ca), is proposed to achieve this self-powered gating process [69]. This autonomous switching is realized by a micro-power voltage comparator and a MOSFET. Driven by the TENG device, the comparator is used to compare the rectified voltage U_T_ with the preset reference voltage U_ref_. When U_T_ is less than U_ref_, the comparator outputs a low level, and the MOSFET is off. When UT exceeds U_ref_, the comparator provides a high level to open the MOSFET for releasing energy rapidly. By presetting U_ref_ according to the open-circuit voltage of the TENG device, it can be realized that the switch is autonomously turned on when the voltage of the TENG device reaches the peak, and turns off when the energy is released, which is in accordance with the sequential switching in Figure 7(Cb). A similar switching method was also applied by Liu et al. in a switchable TENG device shown in Figure 7D [71]. The MOSFET is used as a high-frequency switch to generate current pulses series (Figure 7D). The gating signal is applied by a function generator to accurately control the frequency of current pulses; thus, the whole system is not fully self-powered. However, the performance of high-frequency current pulse generation is demonstrated. As seen in Figure 7D, an 80 Hz gating frequency can still generate clear current pulses. This frequency can meet the requirement for most of the neural stimulation applications such as deep brain stimulations and functional electrical stimulations. As seen in Figure 7D, a higher gating frequency also decreases the amplitude of the current pulses, since the total energy generated by the TENG device is conserved. The next milestone is to realize an automatic high-frequency switch for the generation of current pulses whose amplitude is at a range of several volts, which is suitable for nerve stimulations.

It is noted that there are alternative ways to generate high-frequency current pulses without any switching mechanisms. One of the most conventional ways is by using the grating TENG working in sliding mode [99,100,101,102,103,104,105,106,107,108,109,110,111,112,113,114,115,116,117,118,119,120,121,122,123,124,125,126,127]. By using the proper design for the width and spacing of the grating electrode, high-frequency current pulses can be generated even with slow sliding. Another approach is to find a proper high-frequency power source to drive the TENG devices [128,129]. Bhatia et al. demonstrated a purely mechanical system, in which the frequency of the output current pulses could be up to 70 Hz [128]. A spiral spring was used to store the energy harvested from the environment and to drive a TENG device at the desired frequency by means of an appropriate design of gear train, cam, and flywheel. On the basis of a similar idea, Hinchet et al. proposed a thin-film implantable TENG device which could be transcutaneously driven by ultrasound [129]. By involving ultrasound as the power source, the frequency of the current pulses could be up to 10 kHz. As shown, if a high-frequency operation method, either mechanical or acoustic, is applied to a TENG device, high-frequency current pulses are also obtained, and their amplitude and frequency are even more stable and tunable.

## 5. Summary

In summary, a systematic review of the progress and the perspective for a TENG-based self-powered neuroprosthetics is presented. With proper device optimization, all kinds of nervous systems can be directly stimulated. The perspective of a further developed more sophisticated neuroprosthetic system is proposed, which includes a thin-film TENG device with a biocompatible package, an amplification circuit to enhance the output, and a self-powered high-frequency switch to generate high-frequency current pulses for nerve stimulations. The recent development and progress of each part are reviewed and evaluated. The scenario proposed in Figure 3 depicts the future of this TENG-based self-powered neuroprosthetic system, which is promising for the application of DBS, FES, and VNS.

## Figures and Tables

**Figure 1 micromachines-11-00865-f001:**
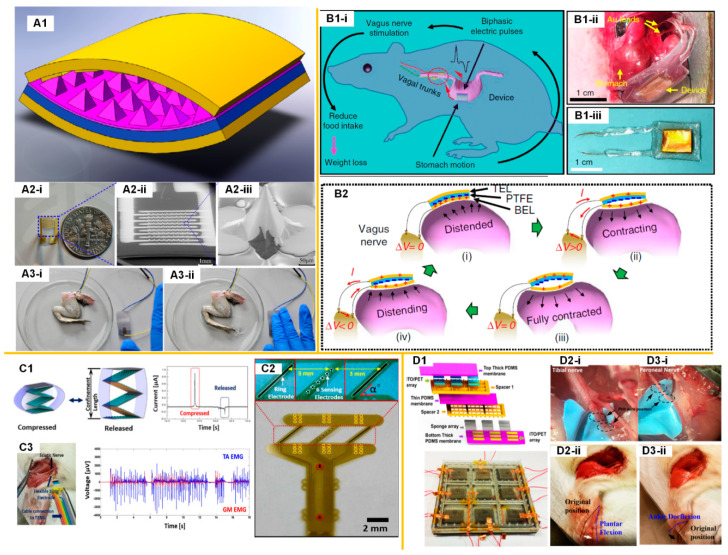
Triboelectric nanogenerator (TENG)-based nerve stimulations. (**A**) The stimulation of a frog’s sciatic nerve using a TENG device connected with a microneedle electrode array (MEA): (**A1**). The 3D structure of the high performance TENG device; (**A2**) The microneedle electrode array used for nerve stimulations; (**A3**) Photos that illustrate the real-time response of frog’s leg by the stimulation of the microneedle electrode array driven by the TENG device. Reproduced with permission from [13]. Copyright 2014 Elsevier; (**B**) The vagus nerve stimulation (VNS) system using a TENG device attach to the stomach [14]: (**B1**) Operation principle of the self-powered VNS system and the implanted TENG device; (**B2**) Schematics of the working principle of TENG device under different stomach motion stages). (**C**) The stimulation of a rat’s sciatic nerve using a stack-layer TENG device connected with a sling electrode: (**C1**) The stack-layer TENG device and its voltage output; (**C2**) The sling neural electrode used for nerve stimulations; (**C3**) The photo that illustrates the sciatic nerve stimulation and recorded EMG signals. Reproduced with permission from [16]. Copyright 2017 Elsevier; (**D**) A selective stimulation of tibial nerve and common peroneal nerve using a multi-pixel TENG device: (**D1**) The multi-pixel TENG device; (**D2**) The plantar flexion induced by the stimulation of the tibial nerve; (**D3**) The ankle dorflexion induced by the stimulation of the peroneal nerve. Reproduced with permission from [15]. Copyright 2018 Elsevier.

**Figure 2 micromachines-11-00865-f002:**
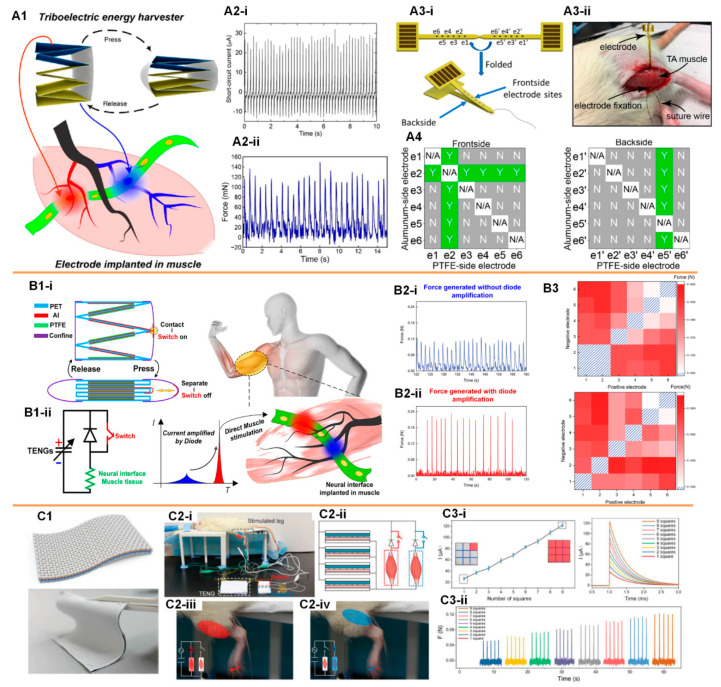
TENG-based direct muscle stimulations. (**A**) The muscle stimulation by a stack-layer TENG device connected with an intramuscular neural probe: (**A1**) Schematics of the self-powered muscle stimulations; (**A2**) The current output of the TENG device (**i**) and the generated force by muscle stimulations (**ii**); (**A3**) The structure (**i**) and the implantation (**ii**) of the intramuscular neural probe used for muscle stimulations; (**A4**) The feasibility of muscle stimulations for different electrode pads combinations. Reproduced with permission from [19]. Copyright 2019 American Chemical Society; (**B**) An improved muscle stimulation method using a switching operation to achieve higher efficiency: (**B1**) Schematics of the switch-involved TENG-based self-powered muscle stimulation; (**B2**) The force of muscle stimulations without (**i**) and with (**ii**) the current amplification; (**B3**) The force of muscle stimulations by different electrode pads combinations. Reproduced with permission from [20]. Copyright 2019 Elsevier; (**C**) According to the switching operation, a soft fabric-based thin-film TENG device can achieve selective and controllable muscle stimulation [21]: (**C1**) The fabric-based thin-film TENG device; (**C2**) Schematics of the selective muscle stimulation by the switch-involved operation of the fabric-based thin-film TENG device; (**C3**) The generated force by muscle stimulations can be controlled by the pressing area of the fabric-based thin-film TENG device.

**Figure 3 micromachines-11-00865-f003:**
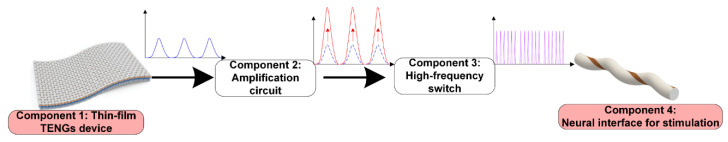
A perspective for the TENG-based self-powered neuroprosthetic system.

**Figure 4 micromachines-11-00865-f004:**
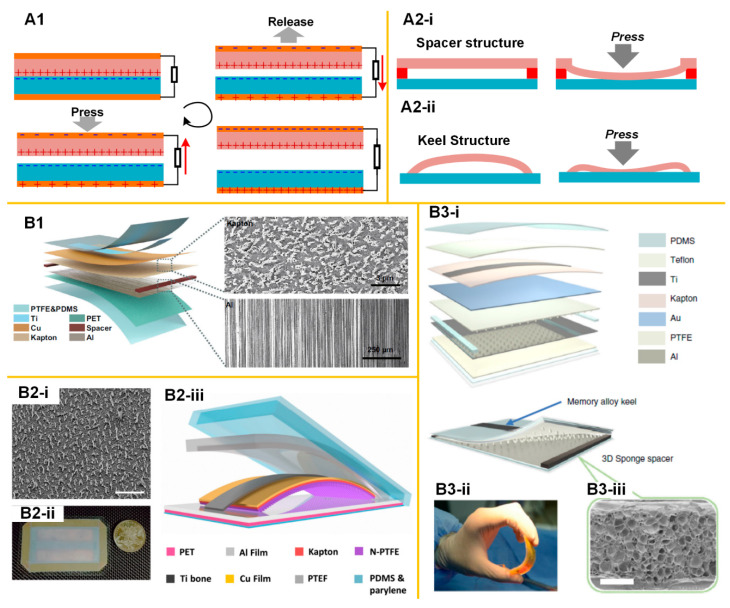
The mechanical structure of implantable thin-film TENG devices. (**A1**) The working principle of the TENG device; (**A2**) The mechanical structure applied for automatic recovery after pressing; (**B1**) The device with spacer and backbone structures [62]; (**B2**) The device with keel structure. Reproduced with permission from [63]. Copyright 2016 American Chemical Society; (**B3**) The device with the integration of both the spacer and keel structures [64].

**Figure 5 micromachines-11-00865-f005:**
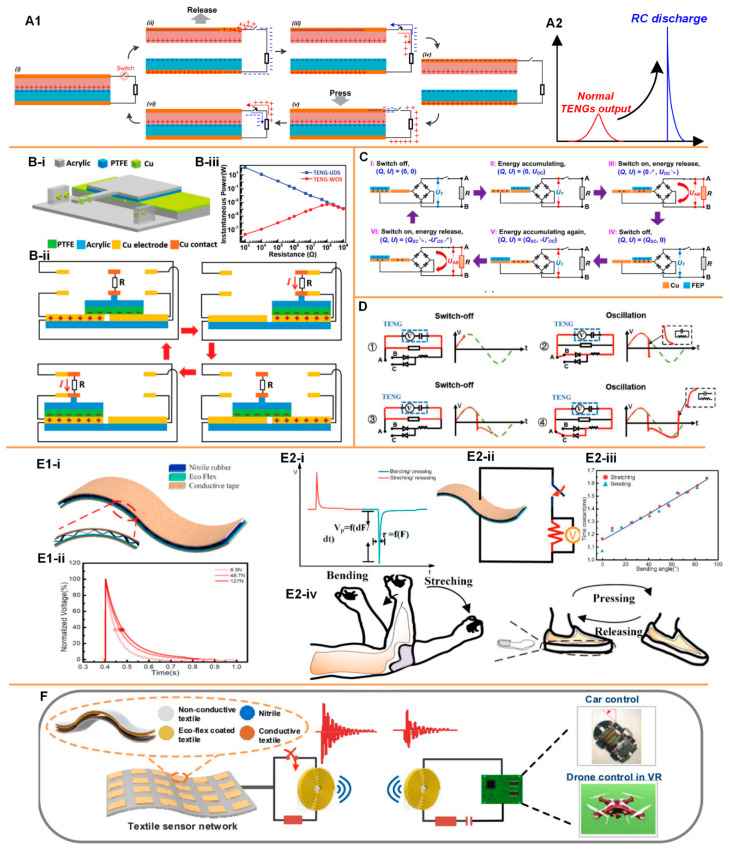
The TENG device integrated with the switching operation. (**A1**) The working mechanism of the TENG device with switching operation; (**A2**) The switching operation can convert the current waveform to an exponential wave (RC discharge); (**B**) A mechanical structure to realize the switching operation automatically during the operation. Reproduced with permission from [65]. Copyright 2018 WILEY-VCH Verlag GmbH & Co. KGaA, Weinheim; (**C**,**D**) Power management methods based on the switchable TENG devices, (**C**) reproduced with permission from [69] and copyright 2017 Elsevier, and (**D**) reproduced with permission from [70] and copyright 2019 WILEY-VCH Verlag GmbH & Co. KGaA, Weinheim; (**E1**,**E2**) Soft thin-film switchable TENG device used for physical sensing: (**E1**) The layer structure of the soft thin-film switchable TENG device (**i**) and its typical output waveform by applying difference forces (**ii**); (**E2**) The application for the physical sensing: (**i**) The physical operations can be decoded from the polarity (positive polarity refers to stretching/releasing and negative polarity refers to bending/pressing) and time constant (bending angle/force); (**ii**) The circuit configuration for the physical sensing; (**iii**) The bending angle decoded from the measured time constant; (**iv**) The application of bending angle and stepping force sensing when the TENG device is attached to the elbow and shoes; Reproduced with permission from [71]. Copyright 2020 Elsevier; (**F**) The Soft thin-film switchable TENG device connected with an inductor for wireless sensing and transmission. Reproduced with permission from [76]. Copyright 2020 Elsevier.

**Figure 6 micromachines-11-00865-f006:**
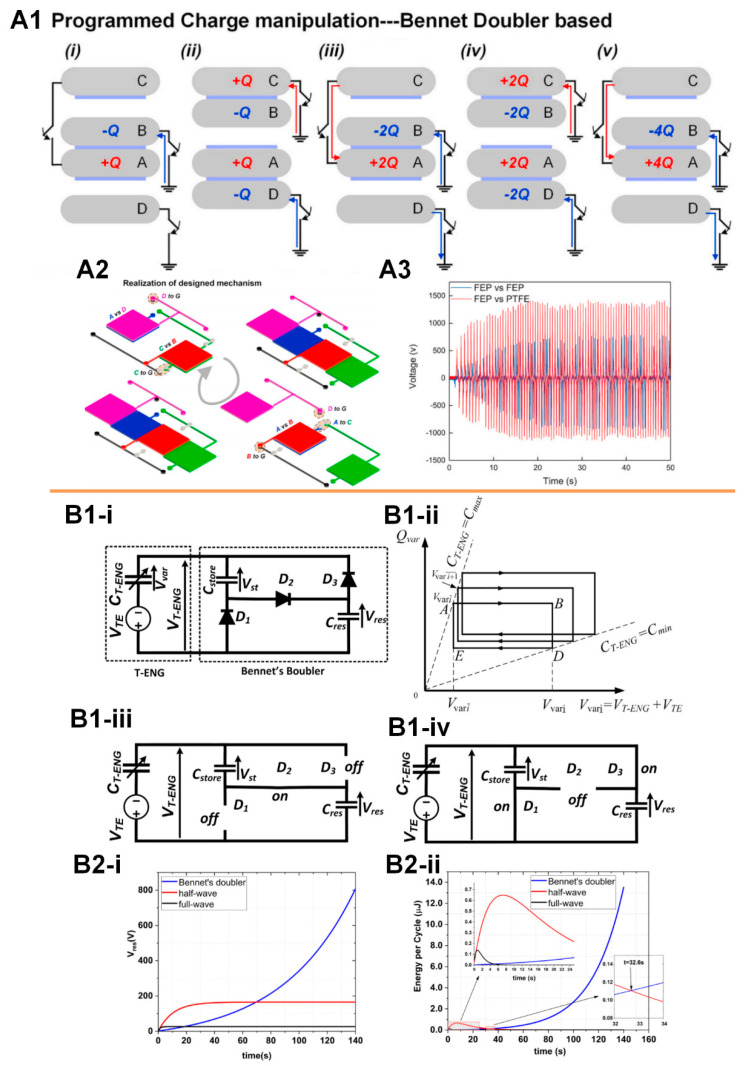
The Bennet’s doubler for the output amplification of TENG devices. (**A**) The mechanism of Bennet’s doubler and its mechanical realization in a TENG device: (**A1**) The mechanism of the bennet doubler based TENG device; (**A2**) The physical realization of the bennet doubler based TENG device in a sliding mode; (**A3**) The typical voltage output of the bennet doubler based TENG device with different surface materials, showing that the amplitude of the voltage gradually saturates at the maximum value. Reproduced with permission from [94]. Copyright 2020 Elsevier; (**B**) The Bennet’s doubler conditioning circuit used for the output power amplification of TENG devices: (**B1**) The principle of the bennet doubler conditioning circuit; (**B2**) The typical output voltage (**i**) and energy (**ii**) of a TENG device with the amplification of the bennet doubler conditioning circuit. Reproduced with permission from [90]. Copyright 2018 Elsevier.

**Figure 7 micromachines-11-00865-f007:**
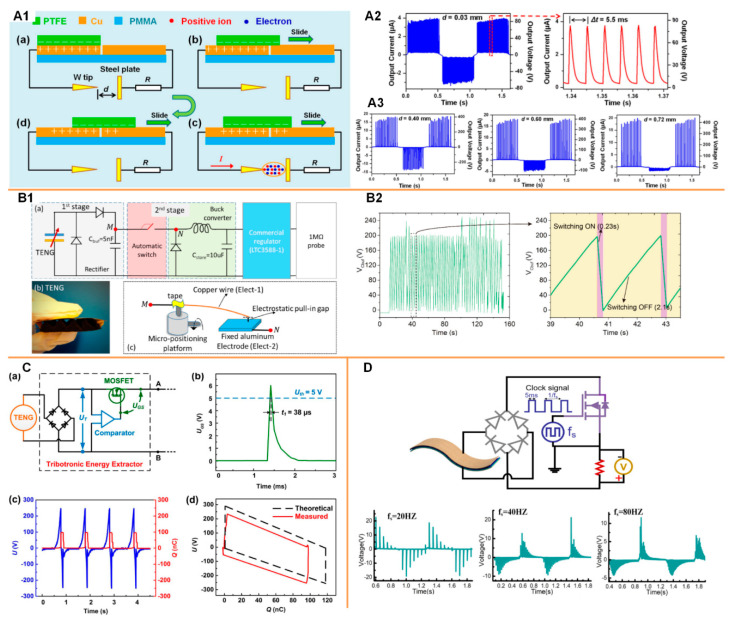
The high-frequency automatic switching methods applied to TENG devices. (**A**) Plasma-based switching method: (**A1**) The plasma generated between the needle-plate by a sliding TENG device; (**A2**) The typical current/voltage of the generated plasma; (**A3**) The asymmetric changing trend of the plasma initiated from the needle (positive) and the plate (negative). Reproduced with permission from [96]. Copyright 2018 Elsevier; (**B**) Electrostatic automatic switching method [97]: (**B1**) (**a**) Schematics of the system configuration for the generation of a stable DC output (**b**) The TENG device used in the system; (**c**) the method of the electrostatic automatic switching; (**B2**) The current pulses generated by the electrostatic automatic switching. (**C**) MOSFET-based auto-switching method. Reproduced with permission from [69]. Copyright 2017 Elsevier; (**D**) MOSFET-based high-frequency switching method applied to the soft thin-film TENG device. Reproduced with permission from [71]. Copyright 2020 Elsevier.

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
