# Peer review of "A Review and Perspective for the Development of Triboelectric Nanogenerator (TENG)-Based Self-Powered Neuroprosthetics"

_micromachines, 2020, doi:10.3390/mi11090865_

Round 1

Reviewer 1 Report

The review paper seems to be interesting and comprehensive at current stage. The manuscript has focused on an aspect or application of triboelectric nanogenerators and has reviewed several articles related to that application. The topic of the review seems to be new and interesting and it is relatively comprehensive and that is the reason for being considered acceptable at the present form.

Author Response

Thanks for the review comments.

Reviewer 2 Report

In this work the authors provide a review of papers focusing on TENG based nerve stimulation. In perspectives they address biocompatibility of the TENG, increasing the TENG current output using switching circuits and Bennet doubler, and increasing the TENG output frequency using MOSFET based circuits.

The paper is well organized and sufficiently covers the latest research. The paper can be accepted after some important corrections.

  1. Since the figures are reproduced from other journals there can be copyright issue. So the authors need to attain the appropriate permissions and provide a statement in all the figures that appropriate permissions were obtained to reproduce the figures.
  2. The authors should provide Section numbering for clarity.
  3. When referring to previous paper the authors use “last author et al.” such as in line number 143, 369, 394, 414, etc. They should correct this to “first author et al.”
  4. Since both high current and high frequency is needed for nerve stimulation, the authors can cite this paper as a potential approach to achieve this: “Bhatia, Divij, et al. "Design of mechanical frequency regulator for predictable uniform power from triboelectric nanogenerators." Advanced Energy Materials 15 (2018): 1702667.”
  5. Minor correction: On line 48 the word “vatious” seems to placed in error.

Author Response

Thanks for review comments.

  1. We have applied and made the statements of all the reproduction permissions for all figures used in the manuscript. 
  2. I have added the section numbers. 
  3. I have revised all cited authors as "first author et al."
  4. Thanks for the suggested paper. We have included in the manuscript and made a brief introduction of the method to achieve high-frequency current pulses. Meanwhile, we notice that there is another paper[1] leverage the similar idea, in which the TENGs is operated by ultrasound rather than a mechanical system, we also include in in this same section. 
  5. We have corrected this typo. Meanwhile, we have double-checked the whole manuscript and confirmed there are no other typos.

In the attachment is the revised manuscript. The revised part is red.

[1] Hinchet, R., Yoon, H.J., Ryu, H., Kim, M.K., Choi, E.K., Kim, D.S. and Kim, S.W., 2019. Transcutaneous ultrasound energy harvesting using capacitive triboelectric technology. Science365(6452), pp.491-494.

Round 2

Reviewer 2 Report

The authors have addressed my concerns in the revised version. Therefore I recommend this paper for publication in Micromachines.